# Single-Nucleus Chromatin Accessibility and Epigenetic Study Uncover Cell States and Transcriptional Regulation of Epidermis in Hidradenitis Suppurativa

**DOI:** 10.3390/biomedicines13071599

**Published:** 2025-06-30

**Authors:** Safiya Haque, Suha Mohiuddin, Jasim Khan, Suhail Muzaffar, Sudeepthi Vejendla, Yanfeng Zhang, Masakazu Kamata, Lin Jin

**Affiliations:** 1Heersink School of Medicine, University of Alabama at Birmingham, Birmingham, AL 35294, USA; 2Department of Dermatology, Heersink School of Medicine, University of Alabama at Birmingham, Birmingham, AL 35294, USA; 3Department of Genetics, University of Alabama at Birmingham, Birmingham, AL 35294, USA; 4Department of Microbiology, Heersink School of Medicine, University of Alabama at Birmingham, Birmingham, AL 35294, USA; 5Center for Epigenomics and Translational Research in Inflammatory Skin Diseases, University of Alabama at Birmingham, Birmingham, AL 35294, USA

**Keywords:** hidradenitis suppurativa, epigenetic regulation, single-nucleus chromatin accessibility, ATF3

## Abstract

**Background/Objectives:** Hidradenitis suppurativa (HS) is a complicated chronic inflammatory skin disorder characterized by recurrent and painful deep-seated nodules, abscesses, fistulae, scarring, and sinus tracts. HS most commonly affects high-density hair follicles and apocrine gland-rich regions of the body, including the axillae, inguinal folds, breasts, and perianal areas. Although genetic predisposition and environmental factors are known to contribute to the development and the severity of HS, the molecular mechanisms of HS are largely unknown. **Methods:** In this study, we employed global epigenetic and genomic data analysis and single-nucleus ATAC-seq (snATAC-seq) to profile the heterogeneity of HS-associated chromatin accessibility and define the underlying disease drivers. We additionally performed high-resolution immunofluorescence staining to confirm a novel candidate regulator. **Results:** We found that multiple skin development modules and molecular signal pathways were epigenetically dysregulated in HS basal CD49f^high^ cells. Importantly, our snATAC-seq revealed a previously unraveled role for a transcription factor, ATF3, in transcriptionally regulating HS-associated genes. We also delineated the specific ATF3 expression pattern across the HS lesional skin. **Conclusions:** We characterize HS-specific epigenetic plasticity and chromatin state at the single-nucleus level and further underscore a possible mechanism for HS pathogenesis.

## 1. Introduction

Hidradenitis suppurativa (HS) is a chronic inflammatory skin disorder characterized by recurrent and painful deep-seated nodules, abscesses, fistulae, sinus tracts, and scarring, most commonly affecting the axillae, inguinal folds, breasts, and perianal areas [1]. This disfiguring condition is accompanied by substantial pain, social embarrassment, and a significantly reduced quality of life. HS has a prevalence of approximately 0.10% in the United States and higher occurrences are noted among women, biracial groups, and African Americans [1,2]. The risk factors for HS include inherited genetic mutations, metabolic syndrome, -particularly obesity-, and smoking. Among these, obesity and smoking have been most strongly associated with increased disease severity., The most common HS comorbid conditions are cardiovascular disease, inflammatory bowel disease, squamous cell carcinoma, and spondylarthritis [2,3].

While significant advancements and endeavors have been made in unraveling the pathogenesis of HS, its exact disease mechanism remains elusive. Histological studies of HS lesions have revealed a complex and heterogeneous pathology, including epidermal hyperplasia, follicular occlusion, dermal fibrosis, immune cell infiltration, and sinus tract formation. TNF-α and interleukin-17 cytokines play critical roles in the disease progression, with their levels positively correlating with severity. Additionally, microbiome alpha diversity is significantly lower in the skin samples of HS patients, which may be secondary to disease biology [4]. These changes highlight the interplay between dysregulated immune responses, epidermal dysfunction, and microbiome alterations within the affected hair follicles and sweat glands.

To better understand inflammatory skin diseases and identify shared or distinct molecular signatures, HS is often compared to psoriasis, another chronic immune-mediated disorder. Although both conditions involve activation of the IL-17 cytokine axis and share similar comorbid conditions—such as Crohn’s disease, metabolic syndrome, and cardiovascular disease—they differ markedly in clinical manifestation, disease course, and therapeutic landscape [5,6]. Psoriasis typically presents as well-demarcated red plaques with silver scale affecting the arms, legs, torso, and scalp [5]. In contrast, HS lesions present primarily in the intertriginous areas as malodorous and purulent nodules and abscesses with sinus tract formation and scarring. Psoriasis also benefits from a more established therapeutic landscape that includes numerous FDA-approved biologics [7]. These differences underscore the need to consider HS as a distinct pathological entity, not simply as a clinical or immunologic analog to psoriasis.

While the role of epigenetic regulation has been well-studied in psoriasis, research into epigenetic dysregulation in HS remains limited. Understanding how epigenetic mechanisms contribute to HS-specific disease biology may provide critical insights into its pathogenesis and uncover new therapeutic targets. Despite the rise in biologic therapies improving outcomes across several inflammatory skin diseases, including psoriasis, there are currently only two FDA-approved biologics for HS, adalimumab and secukinumab, monoclonal antibodies targeting TNF-α and IL-17A, respectively [3,7]. This highlights the urgent need for deeper molecular characterization of HS to guide the development of more effective, disease-specific treatments.

In this study, we defined an altered epigenetic reprogramming in HS progenitor keratinocytes and characterized the heterogeneous genomic states across the HS epidermis at the single-nucleus resolution. Moreover, our snATAC-seq data indicated that ATF3 transcriptionally switches the chromatin-accessible state from the normal to the disease condition. Lastly, we experimentally validated the specific expression features of ATF3 in HS lesional skin.

## 2. Materials and Methods

### 2.1. Human Subjects

The Institutional Review Board of the University of Alabama at Birmingham approved the protocol (IRB-300005449) for obtaining surgically discarded skin tissues from healthy and HS subjects. Surgical excisions from 7 patients with HS (Hurley stage 2 or 3) and 7 healthy cohorts from breast or abdominoplasty reduction surgery were collected (see Appendix A Appendix A for detailed patient information). Among these samples, 2 HS and 2 healthy samples were further used to establish snATAC libraries because they have biologically comparative single-gene signatures (healthy vs. HS conditions), as revealed by our previous study [8].

### 2.2. CUT&RUN Sequencing Data Analysis

For the H3K27ac and H3K4me1 CUT&RUN datasets, fastq files originally deposited in GEO were processed using the CutRunTools2.1 pipeline in default settings (GSE226425: GSM7074757/58/60/61/67/68/69/70) [8]. Peak information from the outputs was used for further analysis. Comparisons between the healthy and diseased samples were made using bedtools (v2.31.0, Salt Lake City, UT, USA). Heatmaps were generated between the cases and controls using deepTools (v3.5.6, Göttingen, Lower Saxony, Germany).

### 2.3. Low-Input ATAC Sequencing Data Analysis

For ATAC-seq datasets, raw sequencing data were downloaded from the GEO database (GSE226425, SRR23681785-89) [8]. We used cutadapt to trim the adapters and Hisat2 to align to the genome (hg38). Samtools (v1.12, Cambridge, MA, USA) were used to convert SAM to BAM files, and PCR duplicates were removed. Bam files were converted to BigWig files, and heatmaps were generated using deepTools (v3.5.6, Göttingen, Lower Saxony, Germany). Macs2 was used to call the peaks in the default setting. An Atlas of cis-regulatory elements was defined as peaks that are merged from the replicates. This Atlas was then annotated to intergenic and intragenic regions using GENCODE v44 as a reference using bedtools (v2.31.0, Salt Lake City, UT, USA). Differential peaks were compared between the case and control samples using bedtools (v2.31.0, Salt Lake City, UT, USA).

### 2.4. Isolation of Epidermal Cells from Healthy and HS Skin

The healthy and HS skin was digested with 2.5U/mL Dispase-II (Millipores Sigma, St. Louis, MO, USA, Cat#4942078001) in Hanks’ Balanced Salt Solution (HBSS, Corning, Corning, NY, USA, Cat#21-022-CM) overnight at 4 °C. After digestion, the epidermis was peeled off and minced. The minced epidermis was incubated with 0.05% trypsin-EDTA at 37 °C for 40 min for further digestion. The suspension was filtered through a 40 μm cell strainer and spun down at 300 g at 4 °C for 5 min. Collected single cells were further applied to establish snATAC libraries.

### 2.5. Establishment of snATAC-seq Libraries and Data Analysis

snATAC-seq was performed following the Single Cell ATAC Reagent Kits V2 kit (10XGenomic, Pleasanton, CA, USA). The nuclei were counted and incubated with a transposase of at least 2000 targeted nuclei before the loading of the Chromium Chip E (PN-2000121). Barcoding was performed in the emulsion (12 cycles) following the Chromium protocol. Libraries were indexed for multiplexing (Chromium i7 Sample Index N, Set A kit PN-3000262). The snATAC-seq data were preprocessed for each sample using the 10x Genomics pipeline (Cell Ranger ATAC v2, Pleasanton, CA, USA), aligning reads to the hg38 reference genome with default settings. Subsequent analyses were conducted following the ArchR package workflow (v1.0.3, Stanford, CA, USA). Briefly, fragment files were imported and merged for the quality control steps, which included doublet removal and assessment of the transcription start site (TSS) enrichment. Dimensionality reduction and batch correction were carried out using the *addIterativeLSI* and *addHarmony* functions with default parameters. Cells were then embedded in the UMAP space using the top 30 Latent Semantic Indexing (LSI) dimensions and grouped into clusters. For peak calling, cells within each cluster were aggregated using the *addGroupCoverages* function. Cluster-specific peaks were identified with the *findMacs2* function, and differential accessibility analysis was performed with *addMarkerFeatures*. Marker peaks for each cluster were extracted using the getMarkers function with the threshold (cutOff = “FDR ≤ 0.01 & Log2FC ≥ 1”).

### 2.6. Transcription Factor Motif Enrichment Analysis

For TF motif enrichment analysis across all the identified peaks, a curated list of TF motifs was obtained from the CIS-BP (Catalog of Inferred Sequence Binding Preferences) database. TF motif deviation z-scores, computed using the *getVarDeviations* function in the ArchR package, were used to assess TF motif enrichment. Based on these scores, TFs of interest were selected and visualized in the UMAP space. In parallel, gene scores, aggregated from peaks located on or near gene loci, were calculated using the *addGeneScoreMatrix* function with default parameters, and the gene scores corresponding to the selected TFs were then extracted and visualized with the *plotEmbedding* function.

### 2.7. Confocal Immunofluorescence (IF) Staining

For IF staining, skin sections were de-paraffinized, rehydrated, then incubated in antigen-unmasking solution according to the manufacturer’s instructions (Vector Laboratories, Burlingame, CA, USA). The sections were placed in a blocking buffer containing 5% normal goat serum in PBST (PBS + 0.4% Triton X-100) for 1h at 37 °C. The sections were then incubated with primary antibodies against proteins for anti-Krt14 (1:400, Biolegend, San Diego, CA, USA, Cat# 906004) and anti-ATF3 (1:50, Santa Cruz, Dallas, TX, USA, Cat#sc-518032) in blocking solution overnight at 4 °C. Sequential staining was conducted for visualization of more than one target in a single specimen. After washing (3 times, 10 min each) with PBST, the sections were re-incubated with the indicated Alexa-Fluor-conjugated anti-goat or anti-mouse secondary antibodies. The sections were fixed in DAPI-containing Vectashield antifade mounting medium (H-1200, Vectorlabs, Burlingame, CA, USA). Lastly, the sections were visualized under a FLUOVIEW FV3000 confocal microscope (Olympus, Tokyo, Japan) equipped with a FV3000 Galvo scan unit and FV3IS-SW (v2.3.2.169, Tokyo, Japan) software.

### 2.8. Data Availability

The processed data of this study have been deposited in the GEO database under the accession code GSE293399.

### 2.9. Statistics

All the data were statistically analyzed by GraphPad Prism 9.0 (San Diego, CA, USA). An unpaired two-tailed Student’s *t* test was used to analyze the differences between the two groups. Data are represented as mean ± standard deviation (SD). A *p* < 0.05 was considered significant unless specified otherwise.

## 3. Results

### 3.1. Histone Modifications Were Reprogrammed in HS Epidermal Progenitor Cells

Our recent findings have shown that epithelial stem cells contribute to the pathogenesis of HS [8]. To assess whether the inflammatory environment in HS affects the global epigenetic landscape of the epidermal basal cells, we reanalyzed our CUT&RUN data to evaluate the profiling of the proximal enhancers (marked by both H3K4me1 and H3K27ac) in the CD49f^high^ epidermal progenitor cells. The CUT&RUN coverage was defined in the 3 Kb region spanning the transcription start site (TSS) of the genes using rank product analysis in each group. Our data revealed a profound rewiring of H3K4me1- and H3K27ac-peak genes within the proximal enhancers under HS vs. healthy conditions (Figure 1A). The HS-associated activated *(n* = 7658) or inactivated (*n* = 19,369) peaks were further characterized by gene ontology (GO). GO analysis revealed that the activated genes in HS were highly enriched for BMP and fibroblast growth factor receptor signaling, lipid metabolism, T cell-mediated adaptive immune response, and keratinocyte proliferation and migration (Figure 1B). For instance, in HS, H3K4me1 and H3K27ac CUT&RUN peaks were gained at the loci of *BMP6* (a BMP signaling member) and *BCL6* (involved in T cell differentiation), respectively (Figure 1C). However, the genes involved in hair follicle and sebaceous gland development, innate immunity against Fc receptor and Gram-positive bacteria, and ERBB4 signaling had a limited presence of H3K4me1 and H3K27ac modifications in HS, as exemplified by the loci containing *COL5A1* (for skin development) and *EPHB1* (involved in ERBB4 signaling) (Figure 1D). Together, these findings indicate that the proximal cis-regulatory regions are heavily involved in the molecular features of HS pathogenesis, and their openness/closure potentially contributes to HS susceptibility by regulating nearby genes.

### 3.2. Open Chromatin Accessibility on the Whole Genome Was Changed in the HS Epidermis

We next sought to assess the open chromatin accessibility at multiple genomic regions in the physical and diseased states. To this end, we remapped the chromatin sites defined on the intergenic and intragenic regions using our published low-input bulk ATAC data from the CD49f ^high^ epidermal cells [8]. We detected a total of 1989 and 5712 unique ATAC-seq peaks in the HS intergenic and intragenic regions, respectively (Figure 2A). In the healthy condition, 4228 and 15,174 unique ATAC tracks were situated in the intergenic and intragenic regions, respectively (Figure 2B). In other words, these regions potentially lose chromatin accessibility recognized by transcriptional regulatory factors due to the disease pathogenesis. Moreover, the most abundant transposable elements (TEs) in the HS intergenic genome were retrotransposons, including short and long interspersed repeat elements (SINEs and LINEs, 28.68% and 23.17% of the total, respectively) and long terminal repeats (LTR, 28.17% of the total) (Figure 2C, Left). The coverage of SINEs, LINEs, and LTRs in the HS intragenic regions accounted for 33.15%, 25.24%, and 17.68%, respectively (Figure 2C, Right). A similar distribution of TEs was also observed across the healthy whole genome (Figure 2D). Overall, these data suggest that retrotransposons may influence the expression of nearby or distant HS genes by enhancer-like regulatory loops rather than by increasing their copy number within the genome.

### 3.3. snATAC-seq Profiling Revealed the Alteration of Genomic Heterogeneity in the HS Epidermis

To map the chromatin accessibility at the single-nucleus level in HS, we applied the 10X Genomics snATAC-seq technique to isolate nuclei from two healthy and two HS epidermises. After stringent filtering and batch correction, the integrated healthy and HS datasets were visualized using uniform manifold approximation and projection (UMAP) dimension reduction (Figure 3A). We captured 13 chromatin signature clusters according to the aggregation of peak reads to their closest gene loci (up- and downstream 2500 bp of TSS) from two biological replicate snATAC-seq libraries (Figure 3B). These clusters represent nine cell subpopulations across the epidermis, including basal I and II keratinocytes (KTs), basal III and transient KTs, spinous I and II KTs, granular KTs, hair follicle (HF) cells, melanocytes, Langerhans cells, and T cells. Importantly, the snATAC clustering involved key gene patterns consistent with those defined in our previous scRNA-seq study [8], as exemplified by open chromatin regions for *PTTG1* (basal I KT), *KRT5* (spinous KT), *CD207* (Langerhans cell), *CD3G* (T cell), and *SOX9* (HF cell) (Figure 3C). Importantly, we identified snATAC-cluster 8 as being specifically associated with pathophysiological contexts (Figure 3A,B). Unlike other clusters featured by overlapping areas on the combined UMAP across normal and HS conditions, this cluster was distinctly separated from its neighboring cluster 6 (Basal III and transient amplifying keratinocytes) (Figure 3A,B). We determined that this cluster was characterized by early-stage spinous KTs and listed the cluster-specific genes with significantly enriched open chromatin compared with the parallel controls (log2FoldChange > 0.5 and FDR < 0.05, Appendix A Appendix A). These genes are highly related to cytokine receptor–ligand activity-mediated signaling (*GDF10*, *BTC*, *PDGFC*, *IFNK*, *WNT3*, and *TNFSF18*).

To identify the potential transcription factors (TFs) regulating HS-specific gene expression at a single-nucleus level, we used ArchR to quantify certain TF activity and DNA binding motif scores. We focused on two TF families, ATF2/3/4 and NF-κB1/2, previously predicted to be highly enriched in HS-related open chromatin regions based on our bulk ATAC database [8]. Notably, UMAP analysis revealed a distinct increase in the distribution of ATF3 within the HS-specific cluster compared with the healthy control. (Figure 4A,B), suggesting its potential responsiveness in HS gene transcription. We next computed these TF motif scores to quantify their DNA binding specificity along the transition trajectory in HS. As expected, we found that ATF3 shifted the open chromatin-accessible state from the normal to the diseased state (Figure 4C,D, black dot circle for the cluster 8), revealing that the overall chromatin accessibility within the HS gene bodies was potentially driven by ATF3. We also observed that the NF-κB1/2 motifs were predominantly activated in the HS clusters (Figure 4B), consistent with the role of NF-κB signaling in regulating chronic skin inflammation [9,10]. Interestingly, the ATF2/4 binding activity concomitantly increased in the T cell-related gene patterns in HS (Figure 4D, gray dot circle for the cluster 13), raising the possibility of their roles in fueling pathogenetic inflammation in HS.

### 3.4. ATF3 Is Highly Expressed in HS Lesional Skin

Having identified the activated binding feature of ATF3 in the HS epidermis, we sought to examine its protein expression in HS skin. To this end, we performed an immunofluorescent (IF) assay to quantify ATF3-positive cells within the skin samples. Compared to the healthy skin, the lesional area in HS exhibited a significantly expanded number of ATF3+ cells throughout the epithelial layers (Figure 5A1,B1,C, and Appendix A Appendix A). We have previously demonstrated that the proportion of the basal population (KRT14+) in HS skin was significantly greater than that in healthy controls [8]. Compared with the healthy controls, the higher number of ATF3+ cells were not only co-localized with the KRT14+ keratinocytes in the HS basal layer but were also obviously distributed in the multiple non-KRT14+ superbasal layers (Figure 5 B1,C,D), indicating their roles in mediating hyperproliferation and differentiation of the epidermal keratinocytes in the disease progression. Although both healthy and HS dermal cells express ATF3, positive cells were more widely distributed in the HS dermal components (Figure 5A2,B2,E). This increased cell dispersion suggests the role of ATF3 in regulating immune cell and fibroblast activation [11,12,13].

## 4. Discussion

This study represents a global epigenetic and genomic analysis of CD49^high^ keratinocytes within the HS lesional epidermis. GO analysis revealed that genes primed by proximal enhancers in HS progenitor cells were enriched for upregulated BMP signaling and downregulated hair follicle development, respectively. Combining snATAC-seq with spatial localization analysis, we identified that the expression and transcriptional activation of ATF3 were highly associated with the pathogenesis of the disease.

Genetic factors and epigenetic modification play a significant role in regulating the pathophysiology of inflammatory skin diseases (psoriasis, atopic dermatitis, keloids, and hidradenitis suppurativa) [14,15,16,17]. Mutations in *NCSTN*, *PSEN1*, and *PSENEN* have also been reported in sporadic HS from different cohorts of patients (for example, Caucasian, European, and Asian) [18,19]. The expression of several miRNAs (miR-155-5p, miR-223-5p, miR-31-5p, miR-21-5p, and miR-146a-5p) was overexpressed in HS lesional skin compared to normal skin [20]. However, limited research has been conducted on the detailed epigenetic mechanisms involved in the pathogenesis of HS using disease primary tissues or cells. Through dual-omics profiling and CUT&RUN datasets, we identified numerous HS-associated inflammatory enhancers that are epigenetically activated and coexist with the transcriptional states of their target genes in HS CD49f^high^ cells [8]. In the present study, we further identified numerous activated and inactivated proximal enhancers within the HS genes (Figure 1). We reasoned that these epigenetic changes occurred due to unknown genetic variants on the HS gene loci, driving disease susceptibility and the complex pathogenetic traits, including dysregulated hair follicle development, interfollicular keratinocyte proliferation, epithelialized tunnels, and a chronic inflammatory condition. Additionally, histone-specific modification signatures may be changed by HS-related epigenetic regulators, namely, writers, erasers, and readers. Further studies are needed to exactly pinpoint the underlying mechanisms. Finally, since we have limited samples for ATAC-seq in the current study, we cannot rule out the possibility that transposon-derived accessible chromatin regions impact the reshaping of the HS genome.

In exploring the functional consequences of enhancer activation in HS, we identified significant enrichment of the genes involved in bone morphogenetic protein (BMP) signaling (Figure 1B,C). BMP signaling, a branch of non-canonical TGF-β pathways, has previously been implicated in epithelial remodeling and immune regulation, particularly in psoriatic skin where BMP7 expression is elevated in keratinocytes and contributes to inflammatory responses through the recruitment of diverse immune cell types including Th17, NK cells, and regulatory T cells [21,22]. Our epigenomic data reveal that several components of the BMP pathway are transcriptionally upregulated in the HS lesional epidermis, suggesting that aberrant BMP activity may similarly drive epithelial perturbation and chronic inflammation in HS.

Moreover, BMP signaling is important for normal hair follicle development [23]. Follicular occlusion caused by infundibular keratosis and hyperplasia of the follicular epithelium is thought to be an early event in the onset of HS, which eventually ruptures, leading to dermal immune cell infiltration and the formation of the epithelized tunnel [24]. We additionally show that HS-associated epigenetic profiling is characterized by hair follicle abnormalities (Figure 1B). Taken together, we reason that targeting BMP signaling may ameliorate the clinical symptoms of HS-associated follicular epithelium hyperplasia and inflammatory responses.

A compelling finding of the current study is the profile of HS-associated chromatin accessibility and key transcription factors at the single-nucleus level. We demonstrate that snATAC-seq can be integrated with scRNA-seq in the annotation of cell populations and can further refine our understanding of functional heterogeneity in the HS nuclei (Figure 3). The cell-type-specific enrichment of TF binding motifs implicates the activation of ATF3 that potentially promotes HS-specific keratinocyte gene expression and drives the cellular transition from physical to diseased conditions. ATF3 is a member of the ATF/cyclic AMP-responsive element-binding protein (CREB) family of transcription factors and is involved in the regulation of immune responses, apoptosis, DNA repair, and oncogenesis [25,26]. ATF3 has been reported to play a role in dermal matrix remodeling and to upregulate pro-inflammatory genes in fibroblasts. For instance, in the full-thickness skin wound model, ATF3 was found to promote dermal fibroblast activation and collagen production for wound healing [13]. However, limited research focuses on examining the expression and activity of ATF3 in the setting of clinical inflammatory epithelial skin. Therefore, our single-nucleus profiling serves as an invaluable resource for dissecting the regulatory mechanisms underlying HS epithelial cell fates, including disease cell-type patterning and transcription kinetics, and further revealing the ATF3-mediated HS gene regulatory network. Of note, we found an increased binding motif signal for ATF2/4 distributed in the T cell populations in the HS epidermis (Figure 4D), suggesting ATFs may have a unique role in resident T cell activation and immune responses during the disease progression.

## 5. Conclusions

Our data provide novel information regarding the nucleosome occupancy of genomic regions across the heterogeneous HS epidermal cells. We also present evidence that ATF3 is highly activated across HS epidermal and dermal layers, raising the possibility that it could be a potential driver for pathogenic reprogramming in HS lesional skin. Overall, we present findings that provide new insights into the underlying molecular and cellular processes that cause HS pathologies.

## Figures and Tables

**Figure 1 biomedicines-13-01599-f001:**
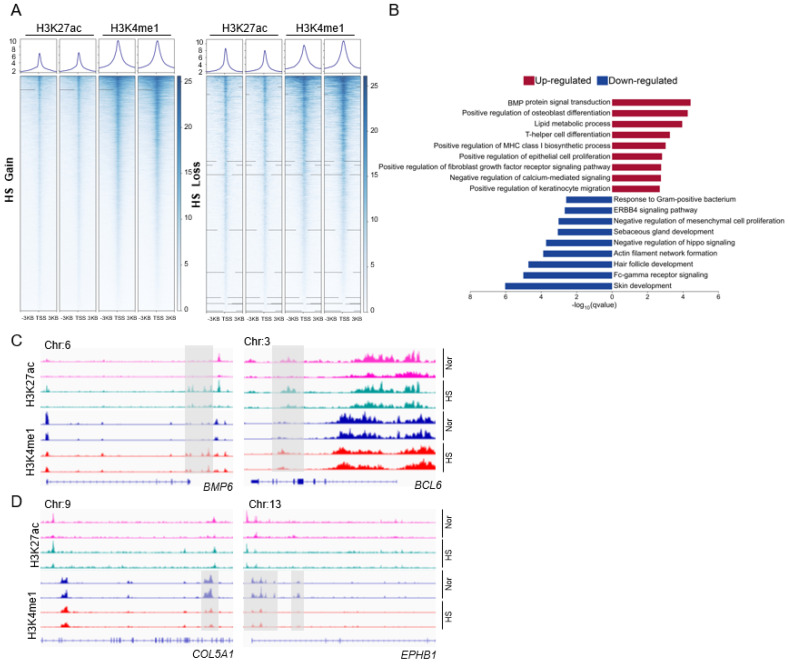
The characterization of the epigenetic regulation between healthy and HS conditions. (**A**) Distribution of gain and loss of indicated histone marks on genomic regions (from −3 kb of TSS to +3kb) in HS CD49f^high^ basal cells (*n* = 2 individual skin). (**B**) GO enrichment analysis showing top-ranked molecular functions and signaling pathways within the genes with proximal enhancers in HS vs. healthy conditions. (**C**,**D**) Genome browser representative views of indicated gene loci characterized by the enrichment of H3K27ac and H3K4me1 CUT&RUN peaks (gray box) in healthy and HS CD49f^high^ epidermal cells. Nor, normal.

**Figure 2 biomedicines-13-01599-f002:**
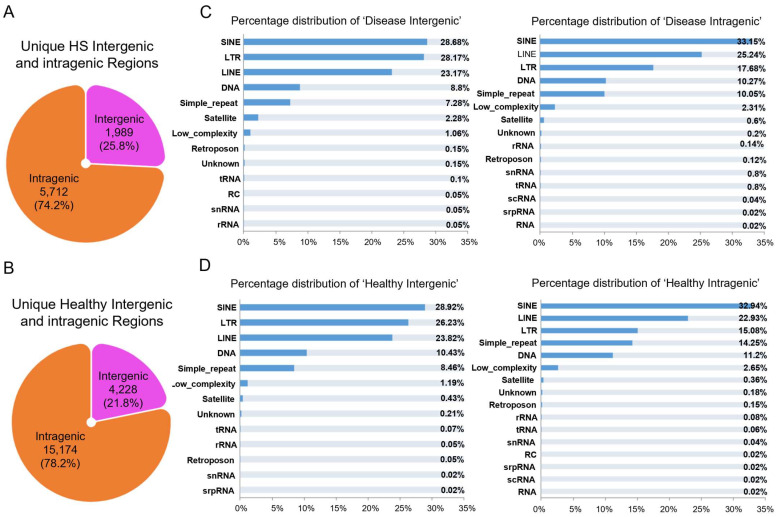
Dynamic changes in genome-wide chromatin state in HS condition. (**A**,**B**) Genomic distribution enrichment of accessible chromatin in the healthy (**A**) and HS CD49f^high^ epidermal cells (**B**) (*n* = 2 individual skin). ATAC peak number and respective occupation percentage of the genome are indicated in the pie diagrams. (**C**,**D**) Transposable elements (TEs) subfamily distribution of accessible TEs in the healthy (**C**) and HS intergenic and intragenic genome (**D**).

**Figure 3 biomedicines-13-01599-f003:**
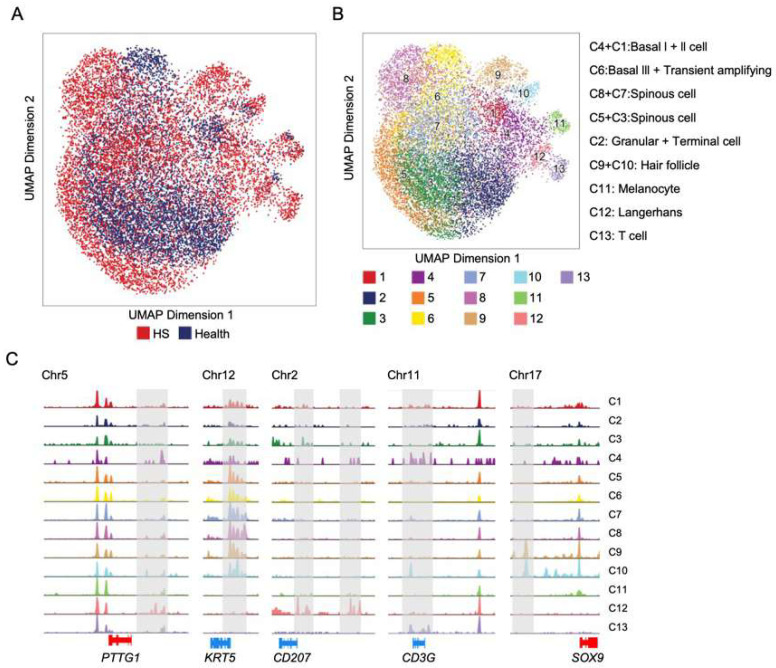
Defining the chromatin accessibility of single epidermal nuclei in healthy and HS lesional skin. (**A**) The integrated snATAC-seq data were visualized with uniform manifold approximation and projection (UMAP) and colored according to unsupervised clustering. Merged snATAC genomics for healthy (*n* = 2 samples) and HS lesional skin (*n* = 2 samples). (**B**) UMAP plot of snATAC-seq dataset with gene activity-based cell-type assignments. Annotated subpopulations are described on the right panel. (**C**) snATAC-seq genomic tracks along denoting chromatin accessibility peaks of marker genes for the representative cluster. Gray boxes highlight specific accessible regions associated with marker genes.

**Figure 4 biomedicines-13-01599-f004:**
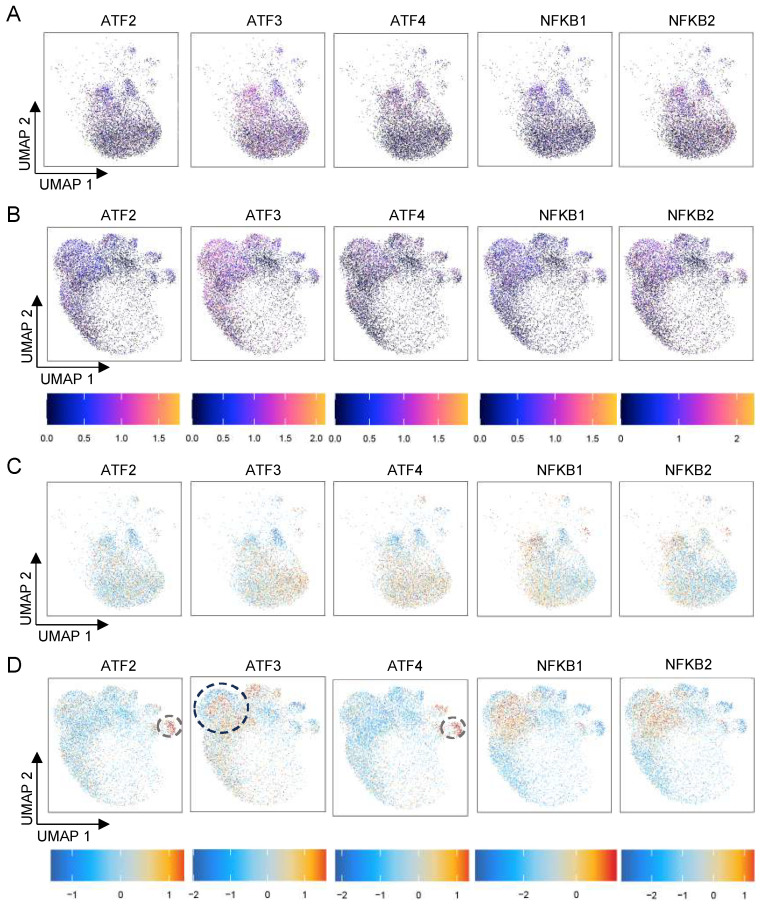
Cell-type-specific transcription factor (TF) activity and its DNA binding score. (**A**,**B**) Indicated TFs overlaid on snATAC-seq UMAP for their activity scores. (**C**,**D**) Predicted DNA binding motif scores for individual TFs from (**A**,**B**). The TF activity and binding motif activity levels are presented with color intensities. A and C, for healthy skin; B and D, for HS skin.

**Figure 5 biomedicines-13-01599-f005:**
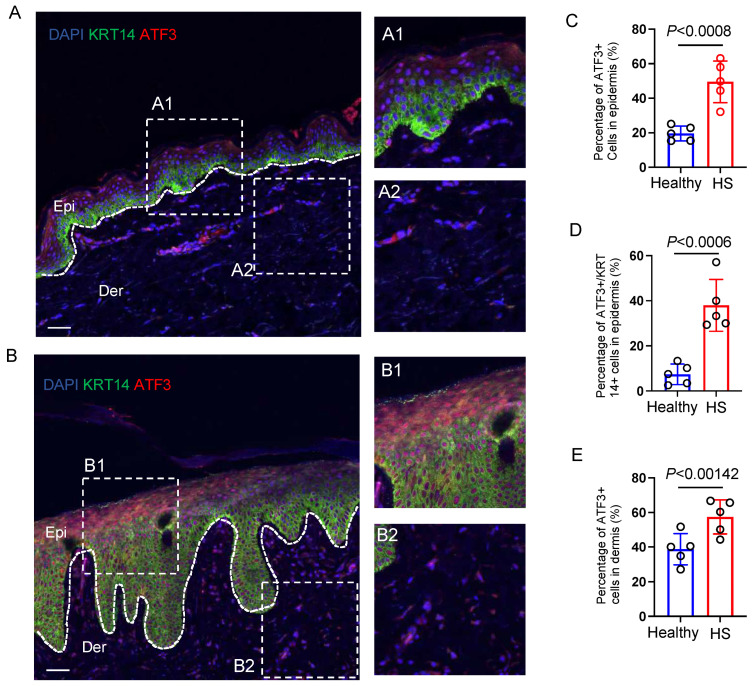
Identification of ATF3 expression in healthy and HS lesional skin. (**A**,**B**) High-resolution images of healthy (**A**) and HS skin biopsies (**B**) immunostained for ATF3. White dot box, representative images with high magnification of healthy and HS lesional tissue sections are depicted. Zoom area from A1, A2, B1, and B2 (left panel) represent the right panel. (**C**–**E**) The percentage of the indicated number of ATF3+ cells across the epidermal layer (**C**,**D**) or the dermal layer (**E**) is determined from histological sections of the respective skin. The dashed lines demarcate epidermal–dermal boundaries. The experiment was performed with five skin tissues each from healthy and HS individuals, respectively, with similar results. Magnification, 20×. Scale bars, 50 μm. Epi, epithelium. Der, dermis.

## Data Availability

The processed data of this study have been deposited in the GEO database under the accession code GSE293399.

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
