# Peer review of "Single-Nucleus Chromatin Accessibility and Epigenetic Study Uncover Cell States and Transcriptional Regulation of Epidermis in Hidradenitis Suppurativa"

_biomedicines, 2025, doi:10.3390/biomedicines13071599_

Round 1
Reviewer 1 Report
Comments and Suggestions for Authors
The present study by Haque et al. investigates the accessibility of chromatin and the underlying regulation in the pathogenesis of HS. The authors use a promising approach and were able to identify ATF3 as a potential key factor in HS.
However, there is still room for improvement in a few areas:
Line 55: please indicate the direction of the correlation
Lines 55-59: The authors describe a reduced alpha diversity in the skin microbiome of HS patients and cite source 4 (Vural et al. 2021). However, I cannot find anything about microbiome in the said reference. Please comment
Line 60: The thematic transition to psoriasis must be better introduced and the differences between HS and Pso must be better emphasized to get a better differentiation
Line 107: A trypsinization time of the cells of 40 min seems very long to me. How did the authors check that this did not lead to stress-induced reactions of the cells that might have had an influence on the experiment?
Line 215: 2 healthy controls and 2 HS patients were used for the sn-ATACseq. How were the samples selected? Why were exactly these persons taken? Please state selection criteria.
Line 218: The word “cluster” appears twice
Line 227: Please explain the pathophysiological context in more detail
Figure 5: ATF3 appears to be generally more strongly expressed epidermally in HS skin. The suprabasal non-KRT14+ colocalized expression is also significantly stronger here. In my view, more emphasis should be made on this.
Line 314: The thematic transition to BMP needs to be better explained.
Author Response
We would like to thank the reviewers for the constructive critiques of our manuscript. We have addressed
all of the points as recommended and feel that our manuscript is now much improved.
1.Line 55: please indicate the direction of the correlation
We have added the positive correlation in the revised manuscript as the reviewer’s request.
2. Lines 55-59: The authors describe a reduced alpha diversity in the skin microbiome of HS patients and cite source 4 (Vural et al. 2021). However, I cannot find anything about microbiome in the said reference. Please comment
We apologize for this mistake. The updated reference according to this statement has been replaced in the reference list.
3. Line 60: The thematic transition to psoriasis must be better introduced and the differences between HS and Pso must be better emphasized to get a better differentiation
We appreciate the reviewer’s thoughtful feedback. In response, we revised the introduction to more clearly establish the rationale for comparing HS to psoriasis (line 60-71). We now introduce psoriasis as a commonly referenced inflammatory skin disease that shares certain immunologic pathways and comorbidities with HS but differs significantly in clinical presentation, disease course, and therapeutic options. We also clarified that although both diseases involve IL-17 signaling, HS should be considered a distinct pathological entity. These revisions aim to strengthen the thematic transition and better differentiate the two conditions within the context of our study’s focus on HS-specific disease biology.
4. Line 107: A trypsinization time of the cells of 40 min seems very long to me. How did the authors check that this did not lead to stress-induced reactions of the cells that might have had an influence on the experiment?
The enzyme-mediated trypsinizing (0.05% trypsin-EDTA) of epithelial cells is a standard protocol for isolating single cells from the epidermis. The treatment time is usually defined within the range of 30 min to 60 min. In our pilot experiment, we set up 30 min, 40 min, 50 min, and 60 min to do such and found that the digestion time of 40 min would give us the best results to obtain individual cells with integrated cell membranes (marked by trypan-blue negative staining). Compared with 0.25% trypsin, 0.05% trypsin is a more gentle concentration and particularly useful for sensitive cells. We don’t think it could bring epithelial cells into a stress-induced condition. Below is one paper that was recently published in JCI. In their Method, the authors even used 0.25% trypsin/EDTA with 10U/mL DNase I to digest epidermis for 1 hour at 37°C.
Reference 1:
- Straalen et al. Single-cell sequencing reveals Hippo signaling as a driver of fibrosis in hidradenitis suppurativa. J Clin Invest. 2024 Feb 1;134(3):e169225
5. Line 215: 2 healthy controls and 2 HS patients were used for the sn-ATACseq. How were the samples selected? Why were exactly these persons taken? Please state selection criteria.
We apologize for the lack of clarity in our original description. The skin samples (two healthy and two HS patient samples) used in this manuscript have comparative single-gene expression signatures (normal vs HS conditions) as revealed by the single-cell RNA sequencing in our previous study (Jin, et al, PNAS; Reference 8 in the original manuscript). Therefore, we believe these samples are biological replicates and represent the physical and disease conditions in the context of gene transcription and they can be used to explore the genomic regulation by snATAC-seq in HS. We have amended this information in the revised manuscript (line 93-96).
6. Line 218: The word “cluster” appears twice
We apologize for this mistake. The correct sentence is shown on line 230 in the revised manuscript.
7. Line 227: Please explain the pathophysiological context in more detail
We have added more detailed description from line 240 to 242.
8. Figure 5: ATF3 appears to be generally more strongly expressed epidermally in HS skin. The suprabasal non-KRT14+ colocalized expression is also significantly stronger here. In my view, more emphasis should be made on this.
We appreciated the reviewer’s insightful suggestions that are helpful in improving the manuscript. The concerns are being addressed in line 286-290.
9. Line 314: The thematic transition to BMP needs to be better explained.
We have revised the writing accordingly (line 331-333).
Reviewer 2 Report
Comments and Suggestions for Authors
In this study, to investigate the relevance of epigenetic regulation in the pathogenesis in hidradenitis suppurativa (HS), an inflammatory skin disease, the authors performed global epigenetic and genomic data analysis and single-nucleus ATAC-seq. The results indicated that epigenetic dysregulations are involved in the pathogenesis in HS, and particularly ATF3, a transcriptional factor, is newly identified as an important molecule in the epigenetic dysregulation in HS. This is well designed study, and the study was adequately performed. The results are clearly mentioned, and the manuscript was properly prepared. The results in this study provide several novel insights into the pathogenesis in HS. However, I have some minor comments, which are described below.
(1) The term "hidradenitis suppurativa epithelial skin" in the title may not be appropriate, and may be changed to more adequate term.
(2) The section 2.5 for the method for "establishment of snATAC-seq libraries and data analysis" (lines 110-129) may be difficult to be understood by most of the readers, who are not familiar with this method. Therefore, the authors may improve this section by addition of more detailed information and explanations.
(3) The section 2.6 for the method for "transcription factor activity analysis" (lines 130-135) may also be difficult to be understood by most of the readers. Therefore, the authors may improve this section by addition of more detailed information and explanations.
(4) At line 64, "there's" may be changed to "there is".
Author Response
1.The term "hidradenitis suppurativa epithelial skin" in the title may not be appropriate, and may be changed to more adequate term.
We agree with the reviewer and have revised the title in this revision version.
2.The section 2.5 for the method for "establishment of snATAC-seq libraries and data analysis" (lines 110-129) may be difficult to be understood by most of the readers, who are not familiar with this method. Therefore, the authors may improve this section by addition of more detailed information and explanations.
We have modified this description in the revised Methods.
3. The section 2.6 for the method for "transcription factor activity analysis" (lines 130-135) may also be difficult to be understood by most of the readers. Therefore, the authors may improve this section by addition of more detailed information and explanations.
More detailed information has been added in this section.
4. At line 64, "there's" may be changed to "there is".
We have revised the whole paragraph according to #1 Reviewer’s comments.
Round 2
Reviewer 1 Report
Comments and Suggestions for Authors
The manuscript by Haque et al. discusses a very interesting aspect of HS pathogenesis. After revision, all my points were addressed. I have no further comments.